# Ssc-miR-141 Attenuates Hypoxia-Induced Alveolar Type II Epithelial Cell Injury in Tibetan Pigs by Targeting *PDCD4*

**DOI:** 10.3390/genes13122398

**Published:** 2022-12-17

**Authors:** Linna Xu, Haonan Yuan, Zongli Wang, Shengguo Zhao, Yanan Yang

**Affiliations:** 1College of Animal Science and Technology, Gansu Agricultural University, Lanzhou 730030, China; 2Gansu Provincial Animal Husbandry Technology Popularization Station, Lanzhou 730030, China; 3National Animal Husbandry Services, Beijing 100026, China

**Keywords:** hypoxia, ssc-miR-141, ATII cells, *PDCD4*, HEK293T

## Abstract

The Tibetan pig is an endemic economic animal in the plateau region of China, and has a unique adaptation mechanism to the plateau hypoxic environment. Research into microRNAs (miRNAs) involved in the mechanism underlying hypoxia adaptation of Tibetan pig is very limited. Therefore, we isolated alveolar type II epithelial (ATII) cells from the lungs of the Tibetan pig, cultured them in normoxia/hypoxia (21% O_2_; 2% O_2_) for 48 h, and performed high-throughput sequencing analysis. We identified a hypoxic stress-related ssc-miR-141 and predicted its target genes. The target genes of ssc-miR-141 were mainly enriched in mitogen-activated protein kinase (MAPK), autophagy-animal, and Ras signaling pathways. Further, we confirmed that *PDCD4* may serve as the target gene of ssc-miR-141. Real-time quantitative polymerase chain reaction (RT-qPCR) analysis was performed to confirm the expression levels of ssc-miR-141 and *PDCD4*, and a dual-luciferase gene reporter system was used to verify the targeted linkage of ssc-miR-141 to *PDCD4*. The results showed that the expression level of ssc-miR-141 in the hypoxia group was higher than that in the normoxia group, while the expression level of *PDCD4* tended to show the opposite trend and significantly decreased under hypoxia. These findings suggest that ssc-miR-141 is associated with hypoxia adaptation and provide a new insight into the role of miRNAs from ATII cells of Tibetan pig in hypoxia adaptation.

## 1. Introduction

Hypoxia is one of the most important environmental factors in the Qinghai–Tibet plateau that exerts a significant impact on the survival of animals and poses challenges to mammals [1,2,3]. Tibetan pigs are endemic economic animals on the Qinghai–Tibet plateau, and can adapt well to the hypoxic environment [4,5]. Physiologists have shown that Tibetan pigs have evolved adaptations to survive in high-altitude hypoxic environments, such as well-developed lungs, thicker alveolar septa, high density of pulmonary arterioles, and large alveoli [6,7,8]. Therefore, the Tibetan pig is the most suitable animal model to explore the hypoxia adaptation mechanism of plateau animals. As the main site of gas exchange, the alveoli are most vulnerable to the influence of external environment [9]. Alveolar epithelial cells increase the expression of cytokines, chemokines, and adhesion molecules upon exposure to a hypoxic environment, which creates an imbalance in the alveolar environment and contributes to a series of lung diseases such as high altitude pulmonary edema, pulmonary hypertension, and pulmonary fibrosis [10,11]. The surface of alveolar epithelial cells is surrounded by ATI and ATII cells. ATII cells can be transformed into ATI cells, which are responsible for the repair, transformation, and regeneration of alveoli. As progenitor cells in the alveoli, ATII cells play a very important role in the repair of lung injury [12]. Royce et al. showed that exosomes from alveolar epithelial cells alleviated pneumonia and pulmonary fibrosis in chronic allergic airway disease and bleomycin-induced pulmonary fibrosis models [13]. Another study showed that non-transplantable bone marrow mesenchymal stem cells (BMSCs) alleviated lung injury through attenuation of lipopolysaccharide (LPS)-mediated damage to ATII cells [14]. In acute lung injury, ATII cells from young mice adapted to stress by increasing the volume and proliferation rate of intracellular surfactants to reduce inflammatory signals and enhance metabolism [15].

Many recent studies have focused on the relationship between microRNAs (miRNAs) and hypoxia adaptation. miRNAs such as miR-21-5p, miR-200b/C, and miR-21 have been shown to play key roles in hypoxia regulation [16,17,18]. Hypoxia-inducible factor (HIF) regulates the expression of a variety of genes that allow cells to adapt to and survive under hypoxic conditions. miRNAs are shown to be involved in regulation of HIF upstream and downstream signaling pathways; for instance, miR-199a, miR-17-92 cluster, and miR-20b regulate HIF1α [19,20,21], and the expression of miR-107, miR-210, and miR-373 was found to be induced by HIF [22,23,24]. We found that ssc-miR-141 expression was significantly upregulated in ATII cells in Tibetan pigs under normoxia and hypoxia conditions. miR-141-3p is associated with cardiomyocyte apoptosis [25], mesenchymal stem cell senescence [26], and I/R injury in endothelial cells [27]. However, the role and mechanism of action of ssc-miR-141 in ATII cells of the Tibetan pig remain to be elucidated.

*PDCD4*, encoding programmed cell death protein 4, is a member of the apoptotic factor family and a key inducer of apoptosis [28]. *PDCD4* plays a pro-inflammatory role in many inflammatory diseases. For instance, *PDCD4* gene deletion significantly reduced the inflammatory response of the spinal cord in a mouse experimental allergic encephalomyelitis model [29] and significantly reduced LPS-induced inflammatory injury in mice and increased their survival rate as compared to that of wild-type mice [30]. In a high fat-induced obesity mouse model, *PDCD4* deletion reduced inflammation in the adipose tissue [31]. In recent years, more and more studies have focused on the regulation of *PDCD4* by miRNAs. Ma et al. [32]. Found that miR-532 could attenuate hypoxia-induced cardiomyocyte apoptosis by targeting *PDCD4* and Zhou et al. [33] showed that the long-noncoding RNA lncRNA-GAS5 regulated *PDCD4* expression by targeting miR-21 and mediated myocardial infarction-induced cardiomyocyte apoptosis. In another study, miR-145 overexpression protected rats from myocardial infarction by targeting *PDCD4* and consequently reducing apoptosis and mitochondrial stress [34]. Although *PDCD4* plays an important regulatory role in various types of apoptosis, its mechanism of action in hypoxia-induced ATII cells of the Tibetan pig remains unclear. Therefore, this study aimed to verify whether *PDCD4* is indeed a target gene of ssc-miR-141 using a dual-luciferase reporter assay system. We performed functional enrichment analysis to investigate the main biological functions of ssc-miR-141 and *PDCD4*, and evaluated the relevant regulatory mechanisms. Our findings will provide better evidence for understanding the regulation of miRNAs in ATII cells of the Tibetan pig under hypoxic conditions.

## 2. Materials and Methods

### 2.1. Samples

A newborn Tibetan piglet was selected and slaughtered under aseptic conditions after anesthesia to collect the lung tissue. Primary ATII cells were isolated from the lungs according to the method of Yang et al. [35]. Isolated ATII cells were randomly divided into two groups, namely the control group (21% O_2_, normoxia) and experimental group (2% O_2_, hypoxia). Cells were cultured in normoxia (74% N_2_, 5% CO_2_, 21% O_2_) or hypoxia (93% N_2_, 5% CO_2_, 2% O_2_) using a mixed three-gas incubator.

### 2.2. Ssc-miR-141 Target Gene Prediction and Functional Enrichment Analysis

We found significant differences in the expression levels of ssc-miR-141 in ATII cells under normoxic and hypoxic conditions based on our previous miRNA sequence data. To further understand the mechanism of action of ssc-miR-141, we used TargetScan (www.TargetScan.org/,accessed on 1 June 2022), MIREAP (http://www.mireap.org/, accessed on 1 June 2022), and miRanda (http://www.bioinformatics.com.cn/local_miranda_miRNA_target_prediction_120/, accessed on 1 June 2022) online software to predict the target genes of ssc-miR-141. Venny2.1 (https://bioinfogp.cnb.csic.es/tools/venny/index.html/, accessed on 5 June 2022) online software was used to determine their intersections. Finally, we used the DAVID6.8 (https://david.ncifcrf.gov/, accessed on 6 June 2022) software to perform Gene Ontology (GO) and Kyoto Encyclopedia of Genes and Genomes (KEGG) pathway enrichment analysis.

### 2.3. Screening of Target Genes Related to Hypoxia Resistance in ATII Cells

Based on the ssc-miR-141 target genes predicted by the three bioinformatic software tools, we determined the intersection of the target gene set using the Venny 2.1 online software. The previous high-throughput sequencing results were screened in cells treated with different oxygen concentrations. Finally, we randomly selected *PDCD4* as the next step for verification and to further screen out the key target genes in ATII cells from the Tibetan pig involved in hypoxia regulation, in combination with results of enrichment analysis of functional genes.

### 2.4. Real-Time Quantitative Polymerase Chain Reaction (RT-qPCR) Analysis

RNA from ATII cells exposed to different oxygen concentrations was extracted by Trizol, and used to synthesize first-strand cDNA according to the Mir-X miRNA First-Strand Synthesis Kit (AG, Changsha, Hunan, China), using Evo M-MLV RT Kit and gDNA clean cDNA of mRNA was generated for qPCR (Accurate Biology) using Mir-X miRNA qRT PCR SYBR kit (AG, Changsha, Hunan, China) and LightCycler^®^ 480 Instrument II (Roche, Basel, Switzerland).

The mature ssc-miR-141 sequence was obtained from the miRBase database. Upstream primers were designed and the mRQ 3’ universal primer was used downstream; U6 was used as the internal reference gene. The mRNA sequences of *PDCD4* and internal reference *β-actin* were selected from National Center for Biotechnology Information (NCBI; https://www.ncbi.nlm.nih.gov/, accessed on 6 June 2022); primers were designed by Premier 5.0 and Primer-BLAST software and synthesized by Zhongke Yutong (Xian, Shanxi, China) Biotechnology Co., Ltd. The details of the primers are shown in Table 1. Reaction conditions were as follows: pre-denaturation at 95 °C for 30 s; denaturation at 95 °C for 5 s, annealing at 60 °C for 35 s, 40 cycles; storage at 4 °C, and analysis of melting curves after amplification. Three replicates were performed for each sample.

### 2.5. Construction of Recombinant Plasmid

According to the data obtained by high-throughput sequencing, the 3′-UTR sequence and mutated sequence of the *PDCD4* gene were synthesized and cloned into a dual-luciferase reporter gene vector (pmirGLO). *Sac*I and *Xho*I (20 bp) restriction sites were introduced into the 5′ and 3′ ends of the target gene sequence, respectively, and the 3′-UTR fragment of *PDCD4* with an ssc-miR-141–binding site was cloned into pmirGLO (Promega, Madison, WI, USA) (Figure 1). The vector was digested with *Sac*I and *Xho*I to obtain the wild-type construct. To construct the *PDCD4* 3′-UTR dual-luciferase reporter wild-type vector (pmirGLO-PDCD4 3′-UTR) and mutant vector (pmirGLO-mut-PDCD4 3′-UTR), miRNA mimics (ssc-miR-141 mimics) and miRNA negative controls (miRNA NC) were designed and synthesized by Gema Pharmaceutical Technology Co., Ltd. (Shanghai, China). The ssc-miR-141 mimics and NC were co-transfected into the HEK-293T cells and their fluorescence activity was detected by the dual-luciferase reporter gene system. When the transcription of firefly luciferase is blocked, the translation of firefly luciferase protein is inhibited and the fluorescence of firefly decreases; however, the expression of *Renilla* luciferase is unaffected and serves as a normalized internal reference. At this time, firefly luciferase activity/Renilla luciferin. Any decrease in the enzymatic activity value can be used to determine whether the miRNA has a direct regulatory effect on the target gene.

### 2.6. Cell Culture and Transfection

HEK-293T cells were purchased from the Cell Bank of the Chinese Academy of Sciences, and cultured in high-glucose Dulbecco’s modified Eagle’s medium (DMEM) containing 10% fetal bovine serum (FBS). After several passages, well-grown cells were seeded in a 24-well plate at a density of about 1 × 10^4^ cells/well. Transfection was performed using Lipofectamine™ Reagent 2000 (Invitrogen, Waltham, MA, USA) and miRNA mimic as per the manufacturer’s instructions. The mimic NC/ssc-miR-141 mimic and PDCD4-3′-UTR wild-type and mutant recombinant plasmids were co-transfected into HEK-293T cells. There were four groups in the experiment as follows: pmirGLO-PDCD4-WT + NC mimic group, pmirGLO-PDCD4-WT + ssc-miR-141 mimic group, pmirGLO-PDCD4-Mut + NC mimic group, and pmirGLO-PDCD4-Mut + ssc-miR-141 mimic group. Three replicate wells were set for each group of samples.

### 2.7. Dual-Luciferase Reporter Gene Activity Assay

Transfected HEK-293T cells were subjected to the luciferase reporter assay after 48 h. The assay was performed in 96-well plates according to manufacturer’s guidelines (Promega, Madison, WI, USA). *Renilla* luciferase activity was normalized to the corresponding firefly luciferase activity and plotted as a percentage of control. Three biological replicates were evaluated for each treatment.

### 2.8. Statistical Analysis

Statistical analysis was performed on the experimental data using IBM SPSS 21.0. Independent sample *t*-test was used for pairwise comparisons, and Duncan’s multiple comparisons and one-way analysis of variance (ANOVA) were used for comparison between multiple groups. Data were expressed as mean ± standard error. GraphPad Prism 8.0 software was used to draw graphs; *p* < 0.05 indicated significant difference, and *p* < 0.01 indicated extremely significant difference.

## 3. Results

### 3.1. Ssc-miR-141 Target Gene Prediction

In this study, 14,353, 13,022, and 11,294 target genes of ssc-miR-141 were predicted from three online software tools. The online software Venny 2.1 was used to determine their intersection, and a total of 8649 target genes were obtained (Figure 2).

### 3.2. GO and KEGG Pathway Enrichment Analysis of Ssc-miR-141 Target Genes

GO functional enrichment analysis was performed on the 8649 target genes predicted for ssc-miR-141; the results are shown in Figure 3A,B. The target genes of ssc-miR-141 were significantly enriched in immune system process (GO:0002376), metabolic process (GO:0008152), biological regulation (GO:0065007), cellular process (GO:0009987), and other entries at the biological process (BP). These target genes were significantly enriched in molecular functions (MF) such as binding (GO:0005488). At the cellular component (CC) level, the target genes were significantly enriched in cell part (GO:0044464), organelle (GO:0043226), etc. Following the GO functional annotation, we performed KEGG pathway enrichment analysis for the predicted target genes. The target genes of ssc-miR-141 were significantly enriched in mitogen-activated protein kinase (MAPK) signaling pathway (ko04010), autophagy-animal (ko04140), cell adhesion molecules (CAMs) (ko04514), and Ras signaling pathway (ko04014) (Figure 3C,D).

### 3.3. RT-qPCR Analysis

To verify the accuracy of the sequencing results and confirm the reliability of the target gene prediction results, ssc-miR-141 and key target gene expression was verified by RT-qPCR. The expression level of ssc-miR-141 in the hypoxia group was significantly higher than that in the normoxia group (*p* < 0.05) (Figure 4B), consistent with the results of high-throughput sequencing (Figure 4A). The expression of *PDCD4* was significantly downregulated in the hypoxia group as compared to that in the normoxia group (*p* < 0.01) (Figure 4C,D). Therefore, based on the negative regulatory relationship between miRNAs and target genes, we selected *PDCD4* as a possible target gene of ssc-miR-141.

### 3.4. Analysis of Ssc-miR-141–Binding Site in PDCD4 3′-UTR

A dual-luciferase activity assay was performed using 293T cells to verify whether ssc-miR-141 binds to the 3′-UTR of *PDCD4*. The luciferase activity was significantly inhibited in the group transfected with ssc-miR-141 mimics + PDCD4 WT (*p* < 0.01) but not in the group co-transfected with ssc-miR-141 + PDCD4 MUT (*p* > 0.05) (Figure 5). These results suggest that ssc-miR-141 mimics are involved in the regulation of *PDCD4* expression by binding to the predetermined binding site in the 3′-UTR.

## 4. Discussion

While the continuing research on miRNAs has revealed the functions of several new miRNAs, the functions of many miRNAs are still unknown. A miRNA regulates the expression of its corresponding target gene. It is quite complicated and difficult to study the target genes of miRNAs through verification. The action sites of miRNAs include the 3′-UTR, 5′-UTR, and the open-reading frame (ORF) of the target gene. One miRNA may act on different target genes or multiple miRNAs may regulate the expression of the same target gene [36]. Therefore, miRNA target gene verification methods can be divided into bioinformatic software-based prediction and experimental verification methods, which can quickly and efficiently screen miRNA target genes and play a very important role in miRNA research. However, the former is associated with a high false-positive rate. Therefore, a combination of experimental validation is usually warranted to confirm the functions of miRNAs. In this study, we used TargetScan, MIREAP, and miRanda to predict target genes, and narrowed down the results to the 8649 common target genes obtained by the intersection. These genes were subjected to GO and KEGG enrichment analysis, from which *PDCD4* was selected as a candidate target gene. Then, a dual-luciferase reporter analysis system was used for experimental verification. This method can accurately and directly determine the binding site of miRNAs on their target genes. Herein, this strategy provided guidance for the in-depth study of the hypoxia adaptation mechanism of the lungs of the Tibetan pig.

In this study, GO and KEGG enrichment analysis of ssc-miR-141 target gene set was performed. ssc-miR-141 target genes were significantly enriched in biological processes such as immune system processes, metabolic processes, biological regulation, and cellular process. ssc-miR-141 was thought to play a role in the immune response of Tibetan pig ATII cells exposed to hypoxia. KEGG analysis showed that ssc-miR-141 target genes were significantly enriched in MAPK, autophagy animal, Ras, and other pathways signaling related to cell repair and immunity. It is speculated that ssc-miR-141 is involved in apoptosis and immune-inflammatory responses that are mediated by targeting these genes involved in the aforementioned signaling pathways.

The current research on the miR-141 family is mainly focused on inflammation and apoptosis. miR-141-5p can affect cervical cancer cell proliferation and apoptosis by targeting *BTG1* [37]. Li et al. [38] found that miR-141-3p promotes nasopharyngeal carcinoma (NPC) by targeting neoplasm metastasis 1 (NME1). Another study [39] found that downregulation of miR-141-3p expression during hypoxia promoted tube formation, migration, and invasion of human umbilical vein endothelial cells (HUVECs) and inhibited apoptosis by targeting *Notch2*. miR-141 induced the Kelch-like ECH-associated protein 1 (Keap1)/NF-E2 p45-related factor 2 (Nrf2) signaling pathway to promote PC12 cell viability and reduce H/R-induced cell damage by inhibiting apoptosis and alleviating oxidative stress [40]. These studies suggest that the regulation of miR-141 expression in cells may be related to apoptosis and inflammation. miR-141 is speculated to regulate hypoxia-induced apoptosis and inflammatory response in ATII cells from Tibetan pigs, which allows them to adapt to the hypoxic environment. The target genes predicted in this study were significantly enriched in immune-inflammation–related signaling pathways such as the MAPK, autophagy animal, and Ras signaling pathways. A large number of studies have shown that the MAPK signaling pathway plays an important role in hypoxic environment. For instance, the activation of MAPK signaling significantly promotes the survival of cardiomyocytes via inhibition of stress of the endoplasmic network. This study further provides new insights in the molecular mechanism of hypoxia-mediated cardiomyocyte injury [41]. miR-19a could attenuate MAPK signaling pathway activity by targeting *CCL20*, and consequently abrogate H/R-induced cardiomyocyte injury [42]. Gong et al. [43] demonstrated the inhibitory effect of miR-20a on the p38 MAPK/c-Junction N-terminal kinase (JNK) signaling pathway via *TLR4* targeting, which effectively protected cardiomyocytes from H/R injury. Thus, miR-20a serves as an alternative target for alleviating myocardial I/R injury. In a hypoxic environment, a regulatory network formed by the NOTCH, hypoxia, and Ras/MAPK pathways may allow animals to adapt to changes in oxygen concentration during developmental processes [44].

Autophagy is a conserved lysosomal degradation pathway that is involved in eliminating damaged organelles and proteins in response to a variety of pathological processes, thereby avoiding excessive damage and dysfunction in various organs and cells [45]. In mice, the antioxidant effect of autophagy maintains the glomerular endothelial cell barrier under starvation conditions by activating the cellular antioxidant system [46]. Zhang et al. [47] showed that autophagy is an adaptation in chronically hypoxic cells. Sexual metabolic response is necessary to prevent an increase in the reactive oxygen species levels and cell death. Therefore, it is speculated that the target genes significantly enriched in the MAPK and autophagy animal, and Ras signaling pathways may be related to the adaptation of Tibetan pig ATII cells to a hypoxic environment. To further confirm the accuracy of sequencing results and target gene prediction results, we performed RT-qPCR analysis to evaluate expression of ssc-miR-141 and the key target gene *PDCD4*. ssc-miR-141 sequencing results were consistent with RT-qPCR verification results, and the hypoxia group had a significantly higher expression level of miR-141 than the normoxia group. These results indicate that ssc-miR-141 activation negatively regulated the expression of its target mRNA *PDCD4* in ATII cells from Tibetan pigs exposed to hypoxia. The expression level of *PDCD4* in the hypoxia group was significantly lower than that in the normoxia group, and the expression trend among the groups was opposite to that of ssc-miR-141. In the dual-luciferase reporter assay, the luciferase activity of cells co-transfected with ssc-miR-141 mimic and pmirGLO-PDCD4-WT significantly reduced, indicating that ssc-miR-141 can target and bind to *PDCD4*. Some studies have found that *PDCD4* plays an important role in miRNA-mediated anti-cardiomyocyte apoptosis, including miR-21 [31] and miR-532 [32]. The above results indicate that the upregulated expression of ssc-miR-141 inhibited the expression of *PDCD4* and alleviated the damage to Tibetan pig ATII cells caused by the hypoxic environment. It is speculated that the overexpression of ssc-miR-141 can effectively enhance the adaptation of the lungs of the Tibetan pig to a hypoxic environment.

## 5. Conclusions

Ssc-miR-141 expression was significantly upregulated in hypoxic environment, and the 8649 target genes were significantly enriched in signaling pathways such as MAPK, autophagy, and Ras. The dual-luciferase reporter gene assay confirmed that miR-141 could directly target and bind to the 3′-UTR region of *PDCD4*, and there was an obvious negative regulatory relationship between them. Our findings provide evidence for the role of ssc-miR-141 in ATII cells of the Tibetan pig, and will contribute to the understanding of miRNA-mediated gene regulation mechanisms during hypoxic stress.

## Figures and Tables

**Figure 1 genes-13-02398-f001:**
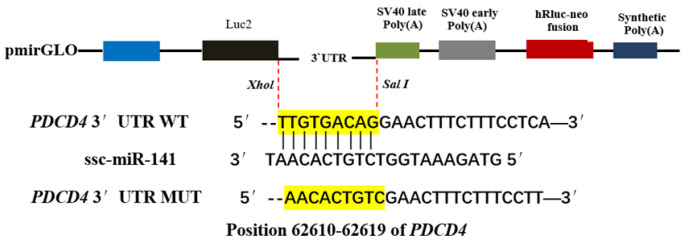
Schematic diagram of pmirGLO-PDCD4-3’UTR recombinant vector.

**Figure 2 genes-13-02398-f002:**
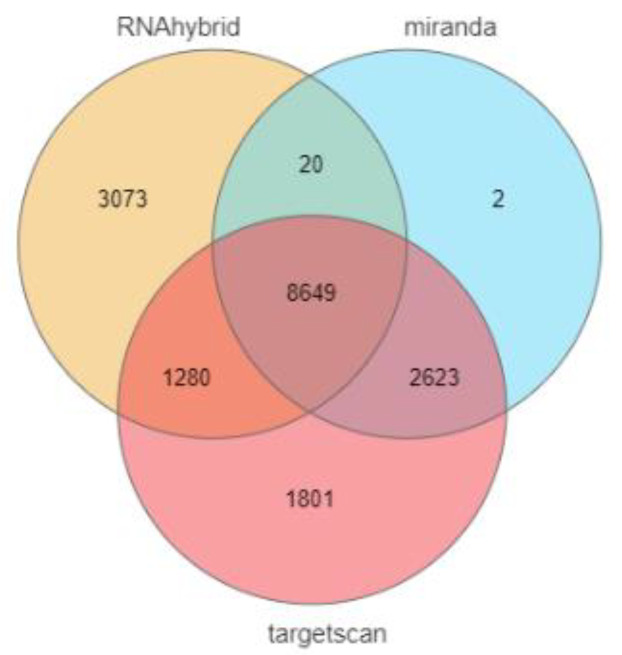
Target gene prediction of ssc-miR-141.

**Figure 3 genes-13-02398-f003:**
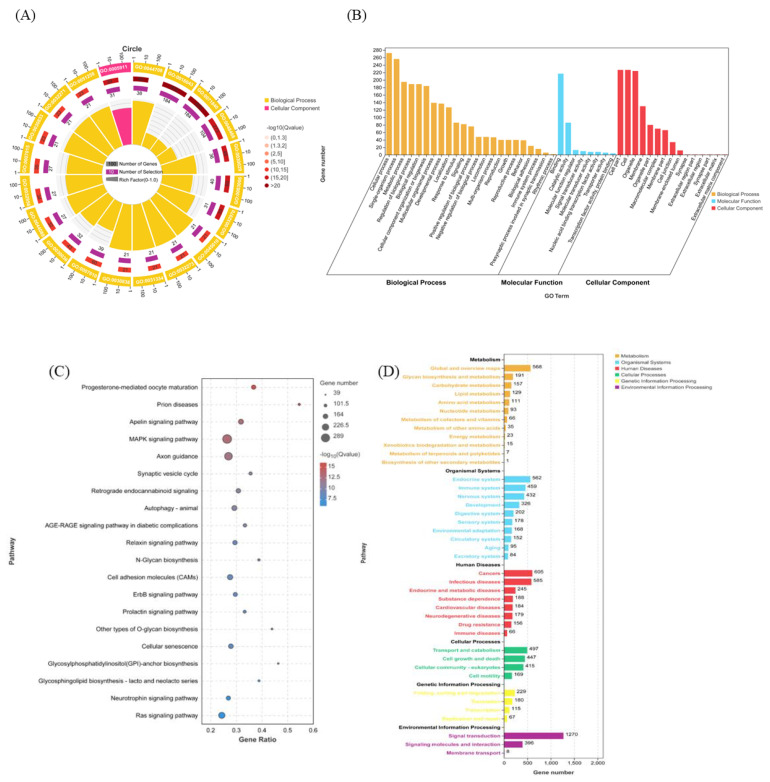
Functional enrichment analysis of ssc-miR-141 target gene sets. GO functional enrichment analysis of ssc-miR-141 target gene (**A**,**B**); KEGG pathway enrichment analysis of ssc-miR-141 target gene (**C**,**D**).

**Figure 4 genes-13-02398-f004:**
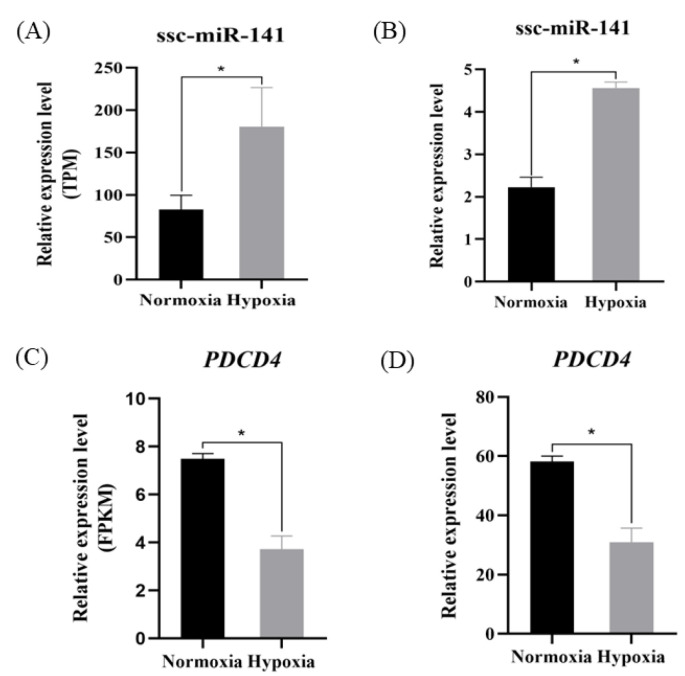
Expression of ssc-miR-141 and *PDCD4* in normoxia and hypoxia groups (ATII cells). (**A**,**C**) indicates the expression of ssc-miR-141, *PDCD4* RNA-seq in normoxia/hypoxia; (**B**,**D**) indicates the expression of ssc-miR-141, *PDCD4* RT-qPCR in normoxia/hypoxia. * indicates *p* < 0.05.

**Figure 5 genes-13-02398-f005:**
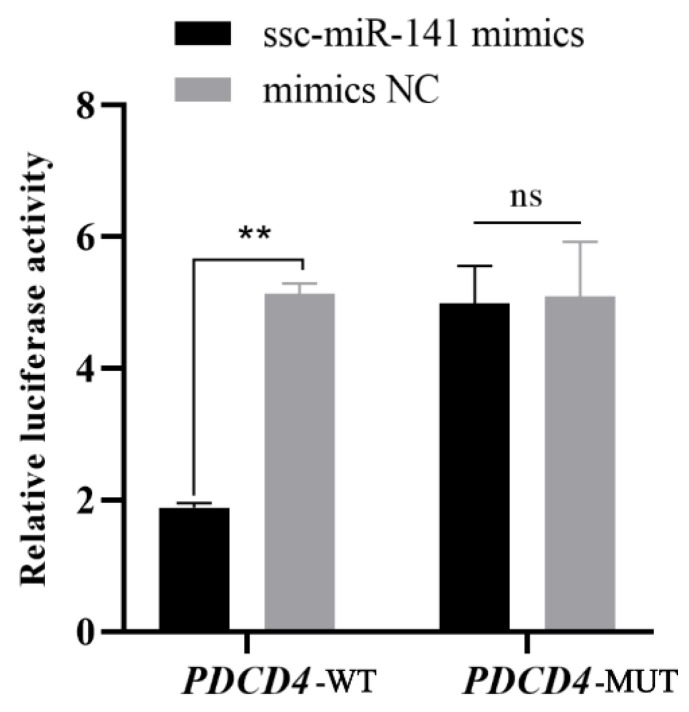
Double luciferase activity assays. Relative luciferase reporter expression was normalized to Negative control(NC). Each experiment was repeated three times. Data are presented as mean ± standard error. ** indicates *p* < 0.01 compared to mock NC; ns indicates no difference compared to mock NC (HEK-293T cells).

**Table 1 genes-13-02398-t001:** Primer information for qPCR.

Gene	Primer Sequence (5′–3′)
ssc-miR-141	Forward: GTAACACTGTCTGGTAAAGATG Reverse: mR Q 3′Primer(Universal downstream primers)
*PDCD4*	Forward: TCATCCCGTGACTCTGGC Reverse: GGTAGTCCCCTTCCTTTCC
*β-actin*	Forward: ATATTGCTGCGCTCGTGGT Reverse: TAGGAGTCCTTCTGGCCCAT
U6	Forward: GGAACGATACAGAGAAGATTAGC Reverse: TGGAACGCTTCACGAATTTGCG

## Data Availability

The following information was supplied regarding data availability: The data is available at NCBI SRA (https://www.ncbi.nlm.nih.gov/sra/, accessed on 20 June 2022) database under accession number PRJNA778032.

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
