# Peer review of "Ssc-miR-141 Attenuates Hypoxia-Induced Alveolar Type II Epithelial Cell Injury in Tibetan Pigs by Targeting PDCD4"

_genes, 2022, doi:10.3390/genes13122398_

Round 1

Reviewer 1 Report

Xu el al., and colleagues have studied hypoxia-induced alveolar type II epithelial cell injury of Tibetan pigs. The authors have tried to establish biological role of ssc-mir-141 and PDCD4 (Programmed cell death protein 4) in the process based on data base analysis. 

The real-time PCR has been done with RNA of ATII cells of Tibetan pig but dual-luciferase reporter assay has been done in 293T, human embryonic kidney cell line. But the reason of using two cell lines of different origin has not been explained in the manuscript. 

The manuscript needs critical reading as there are mistakes in sentence framing.  

Author Response

  1. Reasons why two different sources of cell lines were used
    Dear Editor, Thank you very much for taking your valuable time to review my paper. 293T cells are a human renal epithelial cell line with strong transfection efficiency and value-added rate, which can be well used to demonstrate targeting relationships, but it is only suitable for demonstrating targeting relationships between the two, and for studying subject-related gene functions, while the main effector cells that function in the alveoli are alveolar type II epithelial cells. Therefore, we used two different types of cells.
  2. This piece has errors in sentence structure and needs to be read carefully.
    Dear Editor, I have touched up the article accordingly according to your suggestions, thank you very much for your review.I have uploaded the revised draft as an attachment.

Reviewer 2 Report

1. Excuse me, could the authors please clarify why the gene was chosen randomly? (108) It is not quite clear to me, because the results show a lot of overlap between the three software programs. What about the other genes? 2. I apologize, I would like to have more information about point 2.8 in materials and methods. May the authors tell us what specific data were inputs for the analysis?

Author Response

  1. Excuse me, could the authors please clarify why the gene was chosen randomly? (108) It is not quite clear to me, because the results show a lot of overlap between the three software programs. What about the other genes?
    Dear Editor, Thank you very much for spending your valuable time to review my article. Why we randomly selected the gene PDCD4, we performed GO and KEGG functional enrichment analysis on the common target genes screened by the three software, and our results found that these common target genes were significantly enriched to the signaling pathways of mitogen-activated protein kinase (MAPK), autophagy-animal, and Ras signaling pathways. We then found that these pathways are associated with apoptosis during hypoxia regulation based on our review of the literature, so we screened for the apoptosis-related marker gene PDCD4 in these pathways and determined the regulatory relationship of miR-141 on PDCD4.
  2. I apologize, I would like to have more information about point 2.8 in materials and methods. May the authors tell us what specific data were inputs for the analysis?
    Dear Editor, In the data and analysis of article 2.8, we mainly verified the authenticity of the expression and high throughput data of miR-141 and PDCD4 at different oxygen concentrations, and further we verified the targeting relationship between miR-101 and PDCD4 using a dual luciferase system, and I show you the detailed data in the form of attachments.

Round 2

Reviewer 1 Report

Authors have explained the reason for using two different cell lines in the experiments primarly because of higher transfection efficiency of 293T cells. The real-time PCR shows only the expression level of ssc-miR141 and the target gene PDCD4 under hypoxia and normoxia conditions in ATII cells. But the mechanistic study, which is the crux of the article as the title explains, has been done in 293T which is a human embryonic kidney cell line.  I would suggest to do the functional study in cell line of similar origin for validating the claim because the given effect might not be reproducible in cells of different origin.

OR, the authors should show atleast one or two downstream molecules affected under hypoxia and normoxia in both the cell lines to justify the conclusion.

OR, atleast authors should give some references where such things have been done in two different cell lines. 

The reason should also be included in the discussion section for better understanding of the readers.  

Also, the legend of the figures should show name of the cell lines. 
